# Evidence for suppression of immunity as a driver for genomic introgressions and host range expansion in races of *Albugo candida*, a generalist parasite

Mark McMullan[1,2†], Anastasia Gardiner[1†], Kate Bailey[1], Eric Kemen[3], Ben J Ward[4], Volkan Cevik[1], Alexandre Robert-Seilaniantz[1], Torsten Schultz-Larsen[5], Alexi Balmuth[6], Eric Holub[7], Cock van Oosterhout[4*], Jonathan DG Jones[1*]

[1]The Sainsbury Laboratory, Norwich, United Kingdom; [2]The Genome Analysis Centre, Norwich, United Kingdom; [3]Max Planck Research Group Fungal Biodiversity, Max Planck Institute for Plant Breeding Research, Cologne, Germany; [4]School of Environmental Sciences, University of East Anglia, Norwich, United Kingdom; [5]Department of Plant and Environmental Sciences, University of Copenhagen, Copenhagen, Denmark; [6]J.R. Simplot Company, Boise, United States; [7]Warwick Crop Centre, University of Warwick, School of Life Sciences, Warwick, United Kingdom

**Abstract** How generalist parasites with wide host ranges can evolve is a central question in parasite evolution. *Albugo candida* is an obligate biotrophic parasite that consists of many physiological races that each specialize on distinct Brassicaceae host species. By analyzing genome sequence assemblies of five isolates, we show they represent three races that are genetically diverged by ~1%. Despite this divergence, their genomes are mosaic-like, with ~25% being introgressed from other races. Sequential infection experiments show that infection by adapted races enables subsequent infection of hosts by normally non-infecting races. This facilitates introgression and the exchange of effector repertoires, and may enable the evolution of novel races that can undergo clonal population expansion on new hosts. We discuss recent studies on hybridization in other eukaryotes such as yeast, *Heliconius* butterflies, Darwin's finches, sunflowers and cichlid fishes, and the implications of introgression for pathogen evolution in an agro-ecological environment.

**\*For correspondence:** C.Van-Oosterhout@uea.ac.uk (CO); jonathan.jones@tsl.ac.uk (JDGJ)

†These authors contributed equally to this work

**Competing interests:** The authors declare that no competing interests exist.

**Reviewing editor**: Detlef Weigel, Max Planck Institute for Developmental Biology, Germany

## Introduction

Most parasites have a restricted host range and are often unable to exploit even closely related hosts (*Thompson, 2005*; *Poulin and Keeney, 2014*). Compared to necrotrophs that reproduce on dead plant material, obligate biotrophic parasites can only reproduce on living tissue, and thus are intimately associated with their hosts (*Thines, 2014*). This might be expected to result in host specialization. The adaptive evolution of, for example, new effectors that enable more efficient exploitation of one host species, increases the risk of detection in other host species by triggering their immune system (*Martin and Kamoun, 2012*). Due to this trade-off, the saying 'Jack of all trades, master of none' is especially true for obligate biotrophic parasites, because natural selection that maximises parasite fitness on one particular host species might lead to specialisation and reduced host range (e.g., *Dong et al., 2014*). Yet there are generalist biotrophic parasites that appear to have overcome this evolutionary dilemma and show virulence on diverse hosts.

**eLife digest** Many microorganisms live as parasites inside another living organism, and gain nutrients at their host's expense. Plants and animals have immune systems that serve to protect against this kind of exploitation, but successful parasites have evolved ways to avoid detection by their hosts' immune systems, and/or to suppress hosts' defence mechanisms.

Parasites often avoid detection by releasing molecules that interfere with specific aspects of a host's immune system. The same molecules, however, can be recognised by the immune systems of other species and trigger defence responses that eradicate the parasites; this explains why most parasites can colonise only a limited number of host species. It is less clear how parasites evolve to become 'generalists' that can infect many host species. However, some generalist parasites have several distinct subgroups—each of which is specialised to infect a limited number of host species.

*Albugo candida* is a generalist parasite that infects over 200 plant species, including mustard greens, oilseed and vegetable crops. Even though its looks and its lifestyle resemble those of a fungus, *A. candida* is actually an oomycete: a group of organisms that are more closely related to golden-brown algae than they are to fungi. About 24 subgroups (or 'races') of this generalist parasite have been identified to date, but it remains unclear how these subgroups have evolved.

McMullan, Gardiner et al. tested different isolates of *A. candida*—four from southeast England and one from western Canada—which had been collected from different plant species and confirmed that each could only infect a narrow range of plant hosts.

Next, the genome sequences of these five *A. candida* strains were assembled and compared. This analysis revealed that the five strains represented three distinct subgroups (or races) of *A. candida*. Moreover, some parts of one subgroup's genome were most similar to those found in a second subgroup; and other parts were more like sections of the third subgroup's genome. McMullan, Gardiner et al. point out that such 'mosaic-like genomes' indicate crossbreeding between the different subgroups. But as *A. candida* must infect a plant in order to reproduce, and different subgroups infect different host plants, how can different subgroups meet in order to mate and reproduce?

In answer to this question, McMullan, Gardiner et al. showed that a plant that is infected with one subgroup of *A. candida* becomes susceptible to co-infection with other subgroups, including those that couldn't normally infect this plant species on their own. These findings reveal that generalist parasites can therefore evolve new subgroups via a mechanism that is similar to the way that crossbreeding (or hybridization) between existing species can lead to the evolution of new species.

Some 'generalist' parasite species have solved the dilemma by evolving multiple specialised races, each of which infect different hosts. For example, the eukaryotic oomycete order *Albuginales* consists entirely of obligate biotrophic pathogens that cause disease on a broad range of plant hosts (*Biga, 1955*; *Choi and Priest, 1995*; *Walker and Priest, 2007*). Its largest genus, *Albugo*, comprises ~50 (usually) specialist pathogens (*Choi et al., 2009*; *Thines et al., 2009*; *Ploch et al., 2010*; *Choi et al., 2011*), yet *Albugo candida* (Pers.) Roussel can infect over 200 species of plants in 63 genera from the families of Brassicaceae, Cleomaceae and Capparaceae (*Saharan and Verma, 1992*; *Choi et al., 2009*). *A. candida* infections are the causal agent of 'white blister rust' disease resulting in 9–60% losses on economically important oilseed and vegetable *Brassica* crops. (*Harper and Pittma, 1974*; *Barbetti and Carter, 1986*; *Saharan and Verma, 1992*; *Meena et al., 2002*). *A. candida* consists of different physiological races that each usually show high host specificity (*Hiura, 1930*; *Pound and Williams, 1963*; *Petrie, 1988*). ~24 races of *A. candida* have been defined, based primarily on their host range (*Saharan and Verma, 1992*).

*A. candida* is a diploid organism that reproduces both asexually and sexually (*Holub et al. 1995*), but the relative importance of both reproductive modes is not well established. During asexual reproduction, diploid zoospores are formed in a special propagule (the zoosporangium) on a thallus of hyphae beneath the leaf epidermis. White blisters comprising large numbers of dehydrated sporangia eventually rupture the epidermis to release inoculum for dispersal. When reproducing sexually, fertilization between two isolates results in non-motile diploid thick-walled oospores that can resist extreme temperatures and desiccation. Although the relative importance of different

mechanisms of reproduction in the *Albugo* life cycle remains poorly understood, clonal reproduction enables rapid population expansion, especially in genetically uniform crop monocultures.

*A. candida* is considered a single species despite the fact it comprises several specialised physiological races that colonize different host plants (cf. *Drès and Mallet, 2002*). According to evolutionary and population genetic theory, adaptations and trade-offs associated with host-specialisation combined with strong population structuring can result in adaptive radiation and speciation (*Abbott et al., 2013*; *Stukenbrock, 2013*). Conceivably, *A. candida* is an adaptive radiation 'in progress', and the broad host range is realised by ongoing specialisation of independent races that are on the road to speciation (*Drès and Mallet, 2002*). This raises the important question; does parasite specialization inevitably lead to speciation?

*Albugo* spp. infection strongly suppresses host innate immunity and *Albugo* spp. are unique compared to other microbial plant pathogens in enhancing host susceptibility to secondary infection by otherwise avirulent pathogens, including downy mildews (*Cooper et al., 2008*). It has been suggested that enhanced susceptibility imposed by *Albugo* might accelerate adaptation of other pathogen species to an *Albugo*-susceptible host (*Thines, 2014*). However, no evolutionary rationale has been put forward to explain why it might be adaptive for *Albugo* sp. to render its hosts so susceptible to other pathogens that could compete for access to the same resources (*Cooper et al., 2008*). Arguably, suppression of host innate immunity could facilitate cohabitation of distinct physiological races and thus may enable genetic exchange between them. Introgression, here defined as the introduction by recombination of syntenic nucleotide variation from a parental donor race into the genome of a recipient race (*Hedrick, 2013*), could slow down genetic divergence, and hence, retard speciation. On the other hand, introgression between races that are well-adapted to exploit different host plants could be maladaptive and strongly selected against because hybrids will inherit effector alleles derived from both parental races. Given that immune recognition of even a single effector is sufficient to trigger the immune response and stop an infection, hybrids that possess an expanded repertoire of effector alleles are likely to have a strong fitness disadvantage on most potential host plants.

Much of the ecology and evolution of *A. candida* remains unknown, but with its many specialized races and a broad host range, this 'generalist' plant pathogen is a fascinating study organism. The questions we addressed in the present study are: (1) Are the distinct physiological *A. candida* races genetically isolated and 'on the road to speciation'? (2) Does suppression of host innate immunity enable cohabitation and growth of races with non-overlapping host ranges? To answer these questions, we generated genome sequence assemblies of five isolates that were collected from four host species (*Brassica oleracea*, *Brassica juncea*, *Capsella bursa-pastoris*, and *Arabidopsis thaliana*). We sequenced one isolate of one race, and two isolates of each of two additional races. We show that these races are not genetically isolated despite having non-overlapping host ranges. Recombination analysis shows there is widespread genetic exchange between *A. candida* races, and that hybridisation leading to introgression has occurred numerous times, which include exchanges in the recent past. To explain this observation, we examined whether pre-infection of *Arabidopsis* and *Brassica* with virulent *A. candida* races results in enhanced host susceptibility, and found that pre-infection with a virulent strain enables proliferation of an *A. candida* isolate that would otherwise not colonize that host. For the two races with two isolates, we show that population expansion is by clonal reproduction. We discuss the impact of genetic exchange on *A. candida* evolution, and consider the implications for pathogen evolution and reproduction in an agro-ecological environment.

## Results

### *A. candida* host specificity: single race isolates are host specific

AcNc2 was recovered from infected leaves of *A. thaliana* Eri-1 field-grown plants in Norwich (UK) in 2007. AcEm2 was isolated from wild *C. bursa-pastoris* in Kent (UK) in 1993 (*Borhan et al., 2008*). Isolate AcBoT was harvested from infected inflorescences of *B. oleracea* cultivar 'Bordeaux F1' in Lincolnshire in May 2009, and another isolate AcBoL was harvested from infected leaves of *B. oleracea* in Lincolnshire (UK) in January 2009. Races were single spore purified (*Kemen et al., 2011*). The Ac2V isolate virulent on *B. juncea* was provided by M Borhan (Agriculture and Agri-Food, Canada [*Links et al., 2011*]) and single-spore purified.

**Table 1**. Virulence of the *A. candida* races on different plant host accessions

| A. candida race | Arabidopsis thaliana | | Brassica rapa | | Brassica junceae | | Brassica oleraceae | |
|---|---|---|---|---|---|---|---|---|
| | + | − | + | − | + | − | + | − |
| AcNc2 | 137 | 219 | 0 | 1 | 0 | 1 | 0 | 18 |
| AcBoT | 1* | 34 | 0 | 1 | 0 | 2 | 15 | 0 |
| Ac2V | 1*,† | 107 | 1‡ | 4‡ | 6‡ | 0‡ | 0 | 2 |

\+ Host-pathogen compatible interactions (number of susceptible accessions).

− Host-pathogen incompatible interactions (number of resistant accessions).

*A. thaliana* Ws-*eds1* (<u>e</u>nhanced <u>d</u>isease <u>s</u>usceptibility) mutants were susceptible to all tested *A. candida* races.

†In the laboratory conditions, the cotyledons of the *A. thaliana* accession Ws-3 were found to be susceptible to the Ac2V (**Cooper et al., 2008**).

‡Data from (**Rimmer et al., 2000**) incorporated; in this study, one cultivar *B. rapa* (CrGC1-18, rapid-cycling accession) was infected by Ac2V race and four other tested cultivars ('Torch', 'Colt', 'Horizon', 'Reward') were incompatible with Ac2V race. All analysed cultivars of *B. juncea* (CrGC4-1S, 'Burgonde', 'Domo', 'Cutlass') were susceptible to Ac2V.

We tested the virulence of single race infections of AcNc2, Ac2V and AcBoT on different host species and cultivars. AcNc2 was propagated on *A. thaliana* Ws-2. Two *B. juncea* cultivars ('Cutlass' and 'Czerniac') and 18 cultivars of *B. oleracea* were resistant to AcNc2. We screened 356 *A. thaliana* accessions for their susceptibility to AcNc2 and 38.5% were susceptible (**Table 1**, **Supplementary files 1, 2**). Race Ac2V was originally isolated in Canada on *B. juncea* (**Rimmer et al., 2000**). We confirmed virulence of Ac2V on *B. juncea* by spray inoculation of *B. juncea* 'Cutlass' and 'Czerniac'. On 3-week-old *A. thaliana* plants, all 107 tested accessions were resistant. Two *B. oleracea* cultivars were also resistant to Ac2V (**Table 1**, **Supplementary files 1, 2**). Race AcBoT was virulent on all 15 tested cultivars of *B. oleracea*. In contrast, 34 accessions of *A. thaliana* were fully resistant to AcBoT, as were *Brassica rapa* and *B. juncea* cultivars (**Table 1**, **Supplementary file 1**). These tests confirm that *A. candida* races show pronounced host specificity to distinct host species. While we cannot prove that there are no host species in nature that support growth of more than one of the three races we define here, our experiments and all literature strongly suggest that Ac2V only grows on *B. juncea*, and on some accessions of *B. rapa*, a diploid ancestor of tetraploid *B. juncea* (**Kole et al., 2002**), AcBoT and AcBoL only grow on *B. oleracea*, and AcNc2 and AcEm2 can only grow on a subset of *Arabidopsis* and *Capsella* genotypes. Genetic exchange between races is unlikely to occur unless they colonize the same host. In our study, only the immune-compromised *A. thaliana* Ws-2-*eds1* mutant was susceptible to all races.

## Genome assemblies of *A. candida* isolates

The AcNc2 *A. candida* assembly was used as the reference in this study, and comprises 34 Mb in 5212 contigs of ~160-fold coverage (**Table 2**). We assembled ~73% of an estimated 45 Mb genome of *A. candida* AcNc2 (**Voglmayr and Greilhuber, 1998**; **Links et al., 2011**). In a previous study, a similar proportion (76%) of the *A. candida* Ac2V race genome was assembled (**Links et al., 2011**). The unassembled part of the genome (~11 Mb) is likely to include repeats and duplicated sequences. Repeat sequences constitute ~17.4% of the AcNc2 assembly. Approximately 8% of annotated repeats represent collapsed regions with coverage several times higher than average, so the real repeat content may be higher.

*Ab initio* gene predictions were conducted with several gene prediction programs, resulting in 10,907 predicted gene models. About 90% (9830) of the predicted proteins have homologous sequences in the proteome of *A. candida* Ac2V race (15,824 genes) (**Links et al., 2011**). In about 1000 cases, when we predict a single copy gene in the AcNc2 race, Links and co-authors (2011) have predicted multi-gene families in the Ac2V race, explaining the discrepancy in the predicted gene number for two assemblies.

Only 37% of AcNc2 proteins showed significant sequence similarity to known proteins (BLASTP E-value $\leq 10^{-5}$). Using the Tribe-MCL algorithm, 3522 genes (32% of the predicted AcNc2 gene repertoire) were clustered into 1020 gene families. The largest gene tribes are protein kinases

**Table 2**. Summary of the *A. candida* AcNc2, AcEm2, AcBoT, AcBoL and Ac2V genome assemblies

|  | AcNc2 | AcEm2 | AcBoT | AcBoL | Ac2V |
|---|---|---|---|---|---|
| Number of contigs | 5212 | 11,581 | 11,929 | 11,143 | 12,210 |
| N50 length (bp) | 41,078 | 29,326 | 14,673 | 14,953 | 24,005 |
| N50 number | 231 | 284 | 584 | 581 | 353 |
| Mean contig length (bp) | 6610 | 2668 | 2781 | 2998 | 2770 |
| Assembly size (bp) | 34,454,169 | 33,409,146 | 33,184,526 | 33,409,856 | 33,823,601 |
| GC content (%) | 43.19% | 43.09% | 43.15% | 43.15% | 43.11% |
| CEGMA gene space coverage (%)* | 93.55% | 92.74% | 93.55% | 93,13% | 92.34% |
| Average genome coverage | 160 | 150 | 140 | 140 | 200 |
| Repeat content (%) | 17.4% | NA | NA | NA | NA |
| Predicted genes | 10,907 | NA | NA | NA | NA |

*Completeness of the gene space in the different genome assemblies was estimated using CEGMA pipeline; the presence of over 90% of core eukaryotic genes in the assembly serves as an indication of a overall complete gene space.

(187 members), kinesin- (59 genes) and myosin- (44 genes) like proteins and 35 genes homologous to the secreted 'CHXC' proteins of *Albugo laibachii* (**Kemen et al., 2011**).

915 genes in the AcNc2 assembly are predicted to encode putative proteins with amino-terminal secretory signal peptides, but no trans-membrane domain. Only 34% of the predicted secretome was functionally annotated (BLASTP, E-value $\leq 10^{-5}$), including 115 proteins (proteases, hydrolases, elicitin-like proteins, elicitors, protease inhibitors) that could be involved in plant cell wall degradation and protection against host defense enzymes. In addition to the 35 CHXC proteins (**Tyler et al., 2006**), further candidate virulence factors were identified including 19 homologs of the *Phytophthora* Crinkler effectors (**Haas et al., 2009**), and another 23 secreted proteins with 'RXLR' and 35 proteins with similar 'RXLQ' motifs. Both motifs are located in the N-terminal part of protein after the predicted signal peptide, thus resembling the RXLR effectors of *Phytophthora infestans* and *Hyaloperonospora arabidopsidis* (**Haas et al., 2009**; **Baxter et al., 2010**), but not having any other significant sequence similarity to these proteins.

After carrying out several assemblies based on different k-mer lengths, the quality of each assembly was assessed with various parameters and one best assembly was chosen for each isolate (**Table 2**). The high similarity of the five *A. candida* isolates enabled us to conclude we had sequenced three 'races', within which AcNc2 and AcEm2 were isolates of the same race and AcBoT and AcBoL were also isolates of the same race (**Figure 1**). Therefore, genome comparisons were first conducted on one representative from each race (AcNc2, Ac2V and AcBoT), from each of which ~33–34 Mb of genome was assembled (**Table 2**).

## Genome-wide similarity between races with non-overlapping host range

To assess the overall genome-wide similarity between races, we performed alignments of reads against the AcNc2 reference assembly. For the majority of the AcNc2 genome, we observed a significant positive correlation between read depth in the reference assembly and mapping depth of the Ac2V and AcBoT reads (r = 0.65, p < 2.2e-16; **Figure 2A**). Some AcNc2 regions (3–4% of the assembly) showed low or zero coverage by Ac2V and/or AcBoT reads (**Figure 2—figure supplement 1**), suggesting the presence of highly divergent or unique regions amongst the races. These are gene sparse regions (150 and 234 genes predicted in the AcNc2, respectively), without apparent enrichment for genes encoding for secreted proteins ($\chi 2 = 0.11$, d.f. = 1, p > 0.7). Amplification of randomly selected AcNc2 genes from these regions revealed that four of the selected five genes are indeed absent/or highly diverged in the AcBoT and Ac2V races, and present in the AcNc2 genome.

The overall mean level of nucleotide identity in the homologous genomic regions amongst races is ~99% (**Figure 2B**). We verified 25 polymorphic genomic regions by Sanger sequencing (**Supplementary file 3**). We used the longest of all contigs from AcNc2, 'contig 1' (398,508 bp), to

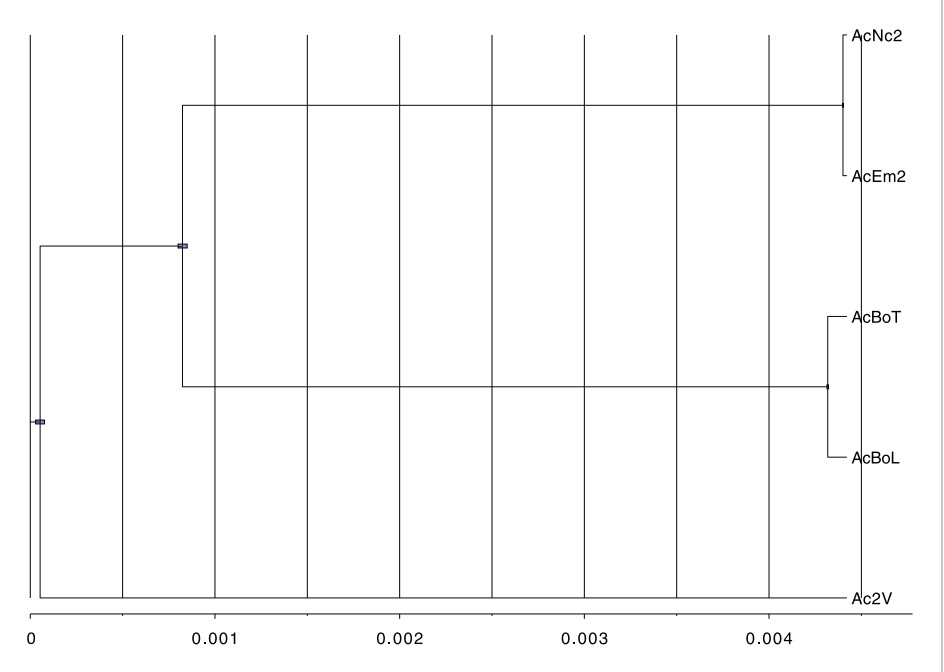

**Figure 1**. Phylogeny shows that the five sequenced *Albugo candida* isolates fall into three divergent races. BEAST tree constructed based on using contig 1 (398,508 bp) shows the divergence among the three *A. candida* races (AcEm2 and AcNc2; AcBoT and AcBoL; Ac2V). Blue bars represent the 95% Higher Posterior Density (HPD). Here, we used a strict molecular clock fixed at 1.0 in order to show the relationship in the scale of substitutions per site.

compare the levels of divergence between races vs the number of heterozygous positions within each race. An extremely low proportion of sites (0.03% and 0.01%) on 'contig 1' are heterozygous within AcNc2, AcEm2 and Ac2V races, respectively (*Figure 3*). Races AcBoT and AcBoL are more heterozygous than Ac2V and AcNc2, with 0.65% of nucleotide positions in 'contig 1' being heterozygous in AcBoT (*Figure 3*). Importantly, >97% of all heterozygous positions are shared in AcBoL and AcBoT (see below). In between-race comparisons, ~1.0% of nucleotide positions on 'contig 1' have diverged between AcBoT, Ac2V and AcNc2.

## Mosaic-like genome structure of *A. candida* races

Polymorphisms are not homogeneously distributed among *A. candida* races. Some regions of the genome are identical for up to 10,000 base pairs, whereas the local nucleotide identity is as low as 89% in other regions of up to 5 kb (*Figure 2B*). We examined 133 contigs (12,373,253 bp), covering 38% of the reference assembly. Stretches of nucleotide similarity amongst races are distributed in a block-like structure; there are regions where AcNc2 is highly similar (or identical) to AcBoT and significantly (nucleotide divergence π > 1%) diverged from Ac2V, and vice versa (*Figure 4*). By using multiple different algorithms that can detect recombination in DNA sequence data incorporated in the software RDP3 (*Martin et al., 2010*), we examined whether this pattern can be explained by genetic introgression amongst races. In addition, we used the software HybRIDS (http://www.norwichresearchpark.com/HybRIDS) to perform a probabilistic recombination analysis, calculate the coalescence time of each recombinant block, and visualize the mosaic-like genome structure.

Recombination analysis of 133 contigs highlighted a total of 675 recombined blocks on 127 contigs that were significant (after Bonferroni correction) in three or more tests using RDP3 (*Supplementary file 4*). The combined length of all identified blocks is nearly 3 Mb or ~25% of the analysed regions. These blocks indicate regions of genome in one race that derive from another race (or the ancestor of another race). *Figure 4* illustrates the effect of genetic introgression on the pattern of nucleotide similarity between the three races in the largest contig of ~400 kb. The sequence (dis)similarity between the three races shows a mosaic-like genome structure with large regions where races AcNc2 and AcBoT show near sequence identity (yellow blocks in *Figure 4B*), whilst other areas show a high

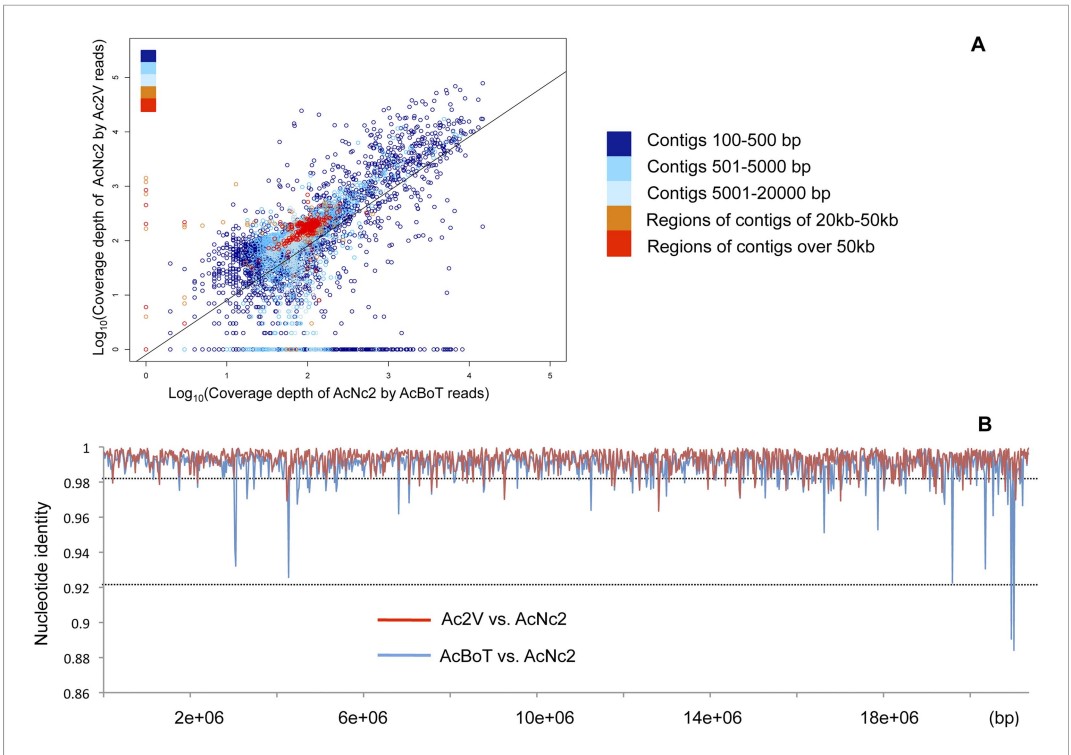

**Figure 2**. Comparison of *A. candida* races using alignments of Illumina reads against the AcNc2 assembly. (**A**) Positive correlation between the depths of coverage of the reference assembly (AcNc2) by the Ac2V and AcBoT reads. For the reference contigs less than 20 kb, the mean coverage was calculated across the whole contig length and log-transformed. For the contigs over 20 kb, the mean coverage was calculated for the sliding window of 20 kb and log-transformed. Y-axis shows the log-transformed depth of the reference coverage by the Ac2V reads; X-axis shows the log-transformed depth of the reference coverage by the AcBoT reads. (**B**) Nucleotide identity amongst the homologous genomic regions of Ac2V, AcBoT and AcNc2. The mean identity was calculated for the sliding window of 20 kb.

The following figure supplement is available for figure 2:

**Figure supplement 1**. Coverage of the reference assembly (AcNc2) by Ac2V and AcBoT.

level of sequence similarity between AcNc2 and Ac2V, and AcBoT and Ac2V (indicated by the purple and turquoise regions, respectively). Note that the presence of such well-defined blocks of high sequence similarity in an otherwise diverged genome is characteristic for rare introgression between organisms that show a high (yet incomplete) level of reproductive isolation. Noteworthy too is the fact that a high level of recombination (relative to the mutation rate) would homogenise the sequence divergence between the races, and hence, that this would not result in the observed mosaic-like structure.

Despite the fact that introgression between races is rare, it must have occurred multiple times between the ancestors of the three races given that the coalescence times varies markedly between the different blocks (*Figure 5*). Assuming a base mutation rate of $\mu = 10^{-8}$ per cell cycle, with 100 cell cycles per year (i.e., a combined mutation rate of $10^{-6}$ per year), analysis in the software HybRIDS show that the most recent introgression event has occurred circa 220 years ago, whilst the oldest event occurred almost 200,000 years ago. The mean age calculated across all introgression events equals 6237 ($\pm$12,594) years (*Figure 5*). (With a combined mutation rate of $\mu = 10^{-7}$ per base per year, the range in the age of introgression would span from 2200 to 2,000,000 years). Irrespective of the mutation rate, the principal finding is that genetic introgression amongst *A. candida* races is an ongoing evolutionary process occurring across a wide range of evolutionary times, and that it gives rise to mosaic genomes with the introgression blocks interspersed in the recipient genomic background.

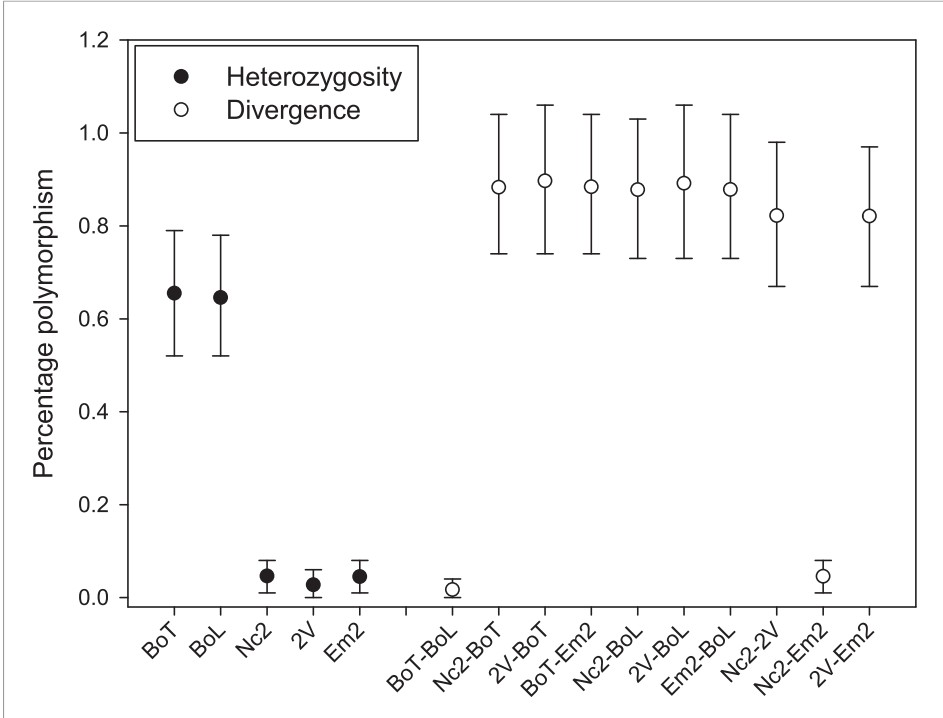

**Figure 3**. Nucleotide polymorphism within and between *A. candida* isolates. Mean (±5–95%CI) polymorphism expressed as the percentage observed heterozygote sites (solid symbols) and percentage nucleotide divergence (open symbols) at contig 1. Confidence intervals were calculated using a bootstrap of contig 1 after removal of indels. Isolates infecting the same host plant (i.e., AcBoT-AcBoL and AcEm2- AcNc2) show little nucleotide divergence, which indicates that they are genotypically almost identical (i.e., diverged by less than 0.05%). Nevertheless, the *Brassica oleracea* infecting race (AcBoT and AcBoL) possess a relatively high heterozygosity compared to the isolates of the *Arabidopsis thaliana* infecting race. Moreover, most of this heterozygous polymorphism is shared (low nucleotide divergence) and presence of the majority of heterozygous sites is consistent with clonal reproduction.

A total of 1655 predicted genes are located in the recombined regions, and amongst these, 125 are predicted to encode secreted proteins. In the introgression regions, we identified 14 genes encoding secreted proteases and hydrolases that in some pathogens act as virulence factors (*Monod et al., 2002*; *Soanes et al., 2007*; *Lebrun et al., 2009*). Thus, recombination between races, resulting in introgression can act as a mechanism for exchange of virulence gene alleles. However, neither gene density nor dN/dS are enriched or depleted in regions affected by introgression, compared to regions not affected (Paired t-test: T = −0.05, p = 0.958; Mann–Whitney test: W = 3.15 × 16⁶, p = 0.152, respectively). It is however entirely likely that by studying only a small subset of all the known races, we have underestimated the actual level of introgression across all races.

## Intra-race diversity suggests clonal propagation after creation of novel adapted allele repertoires

To better understand the evolution and diversification of *A. candida* genomes, we analysed the pattern of recombination and nucleotide divergence with the inclusion of two additional *A. candida* isolates. AcBoL is an additional isolate from the AcBoT race and AcEm2 is an additional isolate of the AcNc2 race (see *Figure 1*). We found evidence for 581 recombination events, of which 335 included AcBoT or AcBoL as recombinants. These two isolates shared a recombinant block in 97.3% of recombination events (AcBoT and AcBoL = 326; AcBoT = 6; AcBoL = 3). AcNc2 and AcEm2 shared a recombinant block in 99.6% of events (AcNc2 and AcEm2 = 246; AcNc2 = 0; AcEm2 = 1). This demonstrates that the AcBoT/AcBoL and AcNc2/AcEm2 races have remained largely unchanged since their initial emergence.

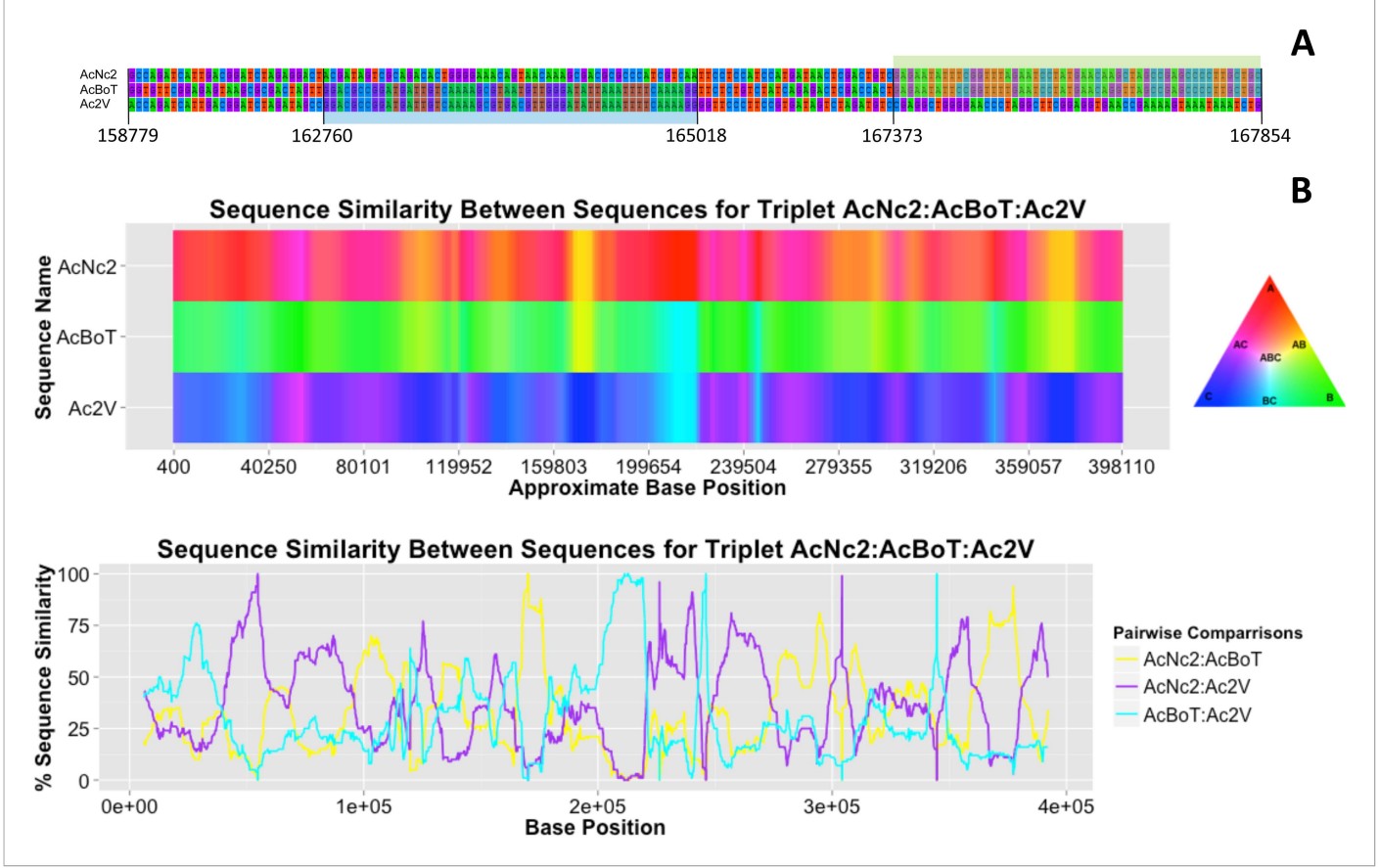

**Figure 4**. Variation in sequence similarity between races. (**A**) Alignment of nucleotides in between positions 158,779 and 167,382 within 'contig 1' of three *A. candida* races (AcNc2, AcBoT and Ac2V) illustrating two recombination blocks coloured blue and green. Both blocks show high sequence similarity between races. Also shown is the sequence divergence in between blocks. Alignment gaps and monomorphic sites have been removed. (**B**) The sequence similarity at 'contig 1' amongst three *A. candida* races was visualised using the colours of a RBG colour triangular in the software HybRIDS (http://www. norwichresearchpark.com/HybRIDS). Areas where two contigs have the same colour (yellow, purple or turquoise) are indicative of two races sharing the same polymorphisms. The linear plot of the proportion of SNPs shared between the three pairwise comparisons between the races. Shown on the X-axis is the actual base position. The graphs were made in the R package HybRIDS (http://www.norwichresearchpark.com/HybRIDS).

The nucleotide diversity within genomes (i.e., the observed heterozygosity) and the nucleotide divergence between genomes (i.e., genetic differentiation) can be used to further understand *A. candida* population biology. Isolates AcBoT and AcBoL were both collected in Lincolnshire in 2009, and the observed heterozygosity in these isolates was at least 13 times higher than that of any of the other races (percentage heterozygous positions in contig 1: AcBoT = 0.653%; AcBoL = 0.644%; AcNc2 = 0.047%; AcEm2 = 0.044%; Ac2V = 0.028%) (see *Figure 3*). Remarkably, AcBoT and AcBoL are heterozygous for almost all of the same sites; 97.2% of the sites that are heterozygous in AcBoT are also heterozygous in AcBoL (and 98.5% vice versa). This is only consistent with clonal reproduction because after just one generation of sexual reproduction (or selfing with recombination), Mendelian segregation would eradicate this high level of genotypic similarity. Notably, such evidence for clonal propagation of a race would be difficult to obtain for haploid fungal pathogens.

With little evidence for recombination and gene flow, most nucleotide divergence between AcBoT and AcBoL must have accumulated through mutation. The nucleotide divergence between these isolates is just 0.030% (121 polymorphisms in 398,508 bp in contig 1) (*Figure 3*). Assuming a mutation rate of $1 \times 10^{-8}$ per cell division, and 100 cell divisions per lineage per year, we estimate that these isolates could have diverged 305 (262–353) years ago. The other pair of isolates, AcNc2 and AcEm2, were collected in Norfolk in 2007 and in Kent in 1993 (160 km apart), respectively. Similar to the

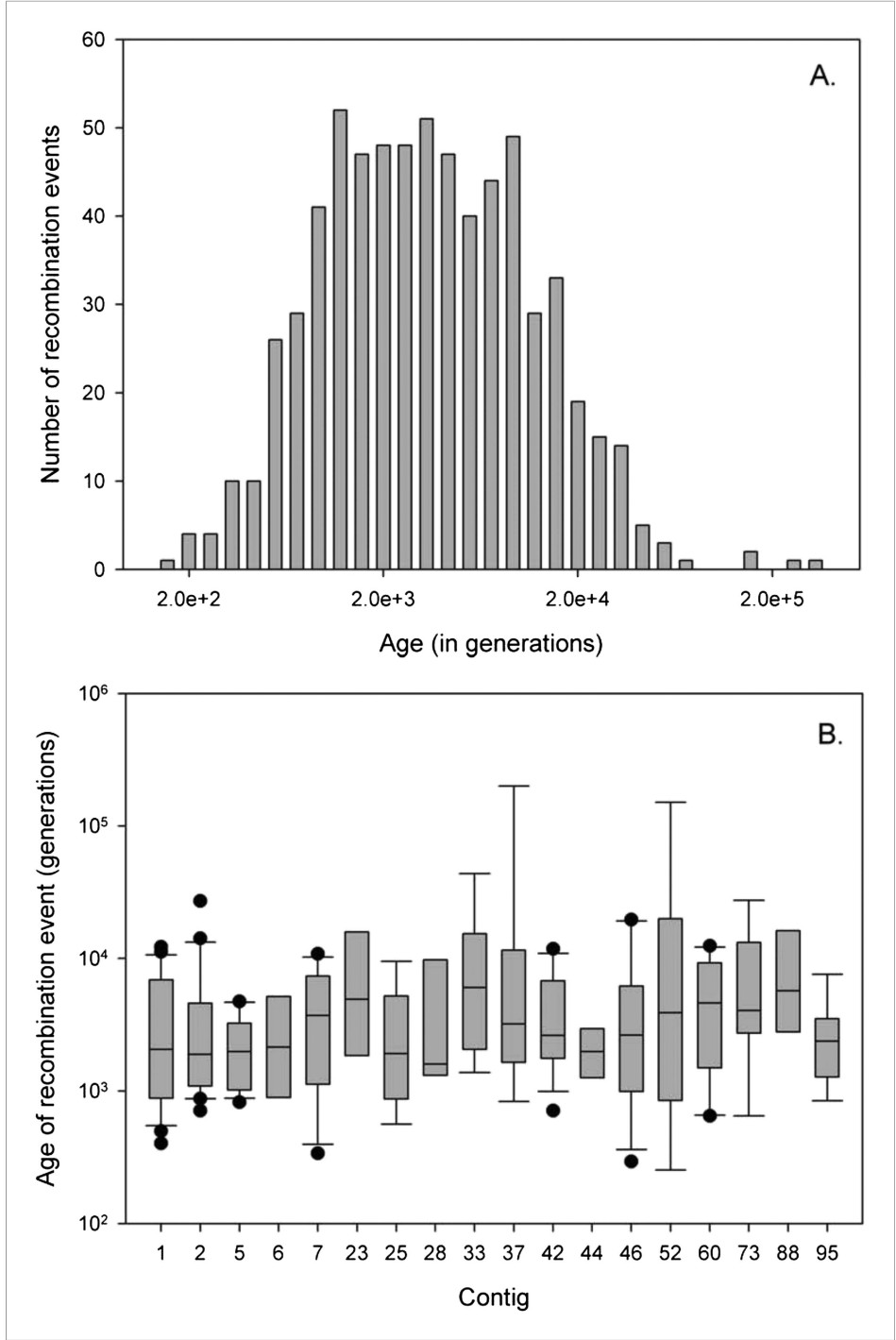

**Figure 5**. Age of recombination blocks. (**A**) Age of the 675 recombination blocks (mutation rate of $\mu = 10^{-6}$) estimated using binomial mass function; (**B**) Boxplot of the median (plus first nation blocks and third quartile) log-age of recombination events in contigs. Only contigs with eight or more events are shown. There is no significant difference in age of events between contigs (GLM: $F_{22, 233} = 1.06$, $p = 0.387$).

former two isolates, AcNc2 and AcEm2 are also nearly identical (nucleotide divergence $\pi = 0.046\%$, that is, they are >99.95% identical) (*Figure 3*). If we again assume that their nucleotide divergence arose by mutation alone, their estimated divergence time is 890 (814–970) years. However, unlike AcBoT and AcBoL, AcNc2 and AcEm2 are much less heterozygous (see *Figure 3*), which suggests that another genetic mechanism, for example, loss of heterozygosity

(*Lamour et al., 2012*), might be operating which has eradicated the gene diversity in the clonally propagating races over time.

## Sequential infections abolish host specificity of susceptibility to *A. candida* infection

*A. candida* infection compromises host resistance against otherwise avirulent pathogen species (*Cooper et al., 2008*). Conceivably, *A. candida* could suppress host defenses to otherwise avirulent races of *A. candida*, enabling co-infection and sexual exchange. To test this we performed sequential inoculation experiments, identifying races using the genome sequences to create race-specific DNA markers. Race-specific PCR of pre-inoculated plants (*Supplementary files 5, 6*) shows preinfection by AcNc2 suppresses resistance in *A. thaliana* accession Ws-2 leaves towards the *B. juncea*-infecting race, Ac2V (*Figure 6A*). Also, preinfection by AcBoT suppresses *B. oleracea* resistance towards Ac2V (*Figure 6B*). Furthermore, defense suppression was so effective that Ac2V was able to complete its life cycle on both *A. thaliana* Ws-2 and *B. oleracea* as observed by successful subsequent infection on *B. juncea* from the sequentially inoculated plants. In a reciprocal experiment, preinfection of *B. juncea* with Ac2V enabled AcNc2 growth on *B. juncea* (*Figure 6C*). Therefore, AcNc2 not only can suppress Ac2V recognition on *A. thaliana*, but Ac2V is also capable of suppressing *B. juncea* resistance towards AcNc2. TIR-NB-LRR resistance genes likely confer Ac2V resistance in *Arabidopsis* (*McHale et al., 2006*), as Ac2V grows on an *eds1-1* mutant of *A. thaliana* Ws-2 (*Supplementary file 6*; [*Parker et al., 1996*]). It has long been noted that *Albugo* sp. have a remarkable capacity to suppress immunity in their hosts (*Cooper et al., 2008*). We hypothesise that suppression of host innate immunity enables co-infection of hosts by races with otherwise non-overlapping host ranges, thus providing a remarkable mechanism to enable sexual genetic exchange between specialised *A. candida* races.

## Discussion

*A. candida* comprises distinct races that specialize on different plant species (*Liu et al., 1996*; *Rimmer et al., 2000*) and its physiological races can infect over 200 species of plants. Here, we describe the genomes of five isolates from three races of *A. candida* that colonize distinct host plant species and appear to have non-overlapping host ranges. Genome analyses show that the races have a mosaic-like genome structure that is consistent with genetic introgression between races that have significantly diverged (mean nucleotide divergence ~1%). Despite this divergence, ~25% of the nucleotide sequence within the analysed genome (3.2 Mb out of 12.4 Mb) is derived from other *A. candida* races. 675 introgressed blocks were identified in the 127 analysed contigs and each block was confirmed by at least three independent recombination algorithms. Based on the high nucleotide similarity of the recombined regions, it appears that some genetic exchange must have occurred recently and that introgression has happened multiple times between the ancestors of the three races.

An alternative hypothesis for the observed mosaic-like genome structure is that the polymorphisms were already present in the ancestor of *A. candida*, and that those may have sorted stochastically among descendant host races. This is known as incomplete lineage sorting (*Pamilo and Nei, 1988*; *Hobolth et al., 2007*), and such ancestrally shared polymorphism is notoriously difficult to distinguish from polymorphisms shared through secondary contact and introgression. However, by estimating the coalescence time of the introgressed regions we showed that hybridisation between the races is an ongoing evolutionary process, with some introgression events occurring as recent as 220 years ago. We therefore reject the hypothesis of incomplete lineage sorting and propose that genetic exchanges between the *A. candida* host races have occurred through rare but nevertheless significant introgression events.

If host-ranges are determined by multiple loci, one might expect that recombination will quickly lead to 'super genotypes' with very wide host ranges. However, this is unlikely because of the dual consequences of effector alleles; they not only can contribute to virulence, but if recognized by *Resistance* (*R*) gene alleles in a particular host, they can result in the complete loss of virulence on that host (*Dodds and Rathjen, 2010*). Effectors selected to facilitate infection and evade recognition by *R* genes in one host might be recognised by the *R* genes in another host and be maladaptive. Thus, genotypes that are highly virulent on multiple hosts are unlikely to arise by recombination between races that specialize on distinct specific hosts, although occasionally, acquiring an

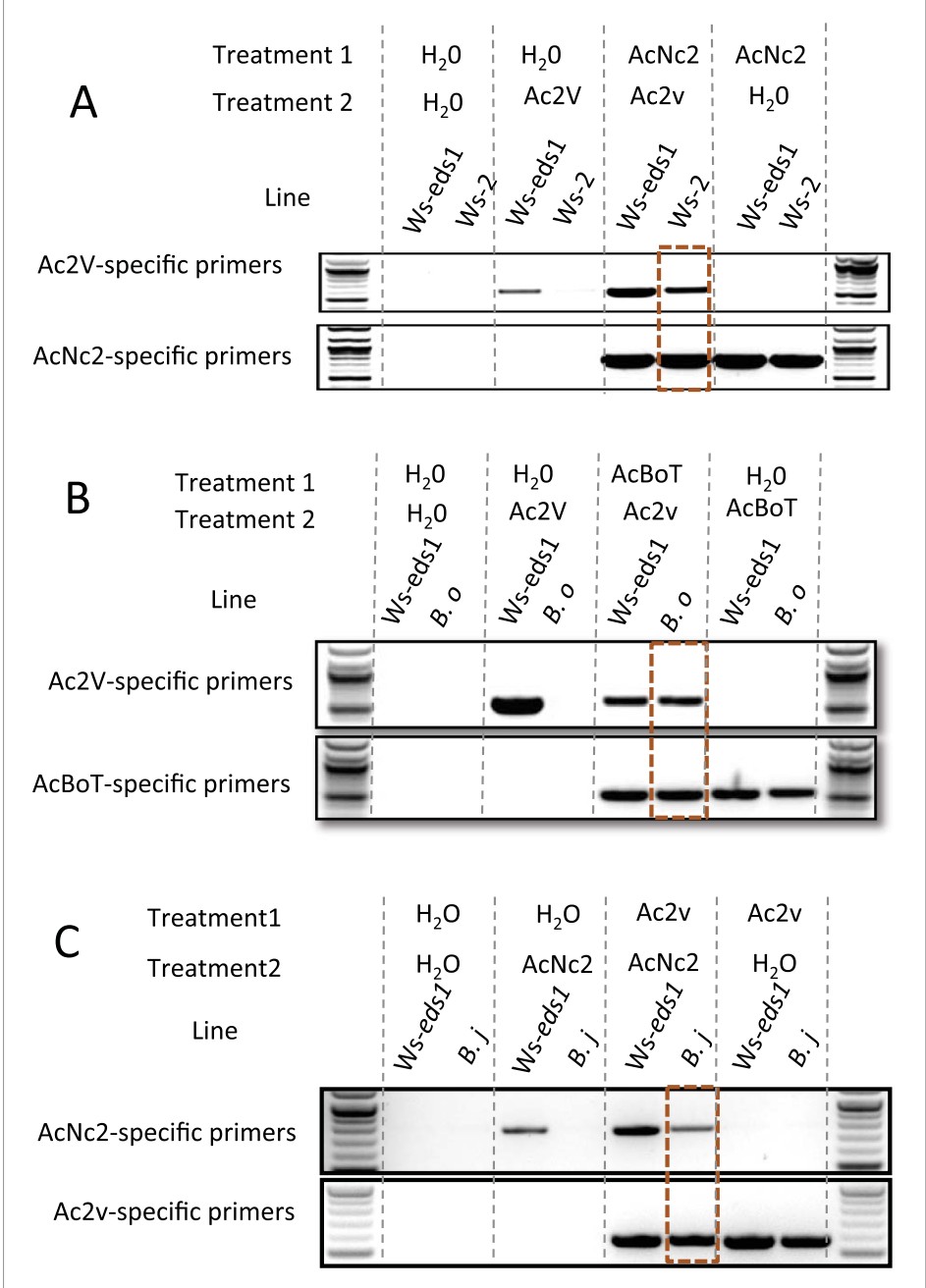

**Figure 6**. PCR with race-specific markers on DNA prepared from plant tissue generated from co-infection assays. (**A**) Co-infection assay of AcNc2 followed by Ac2V onto Ws-*eds1* and Ws-2. (**B**) Co-infection of AcBoT followed by Ac2V onto Ws-*eds1* and *B. oleracea* (*B.o*). (**C**) Co-infection of Ac2V followed by AcNc2 onto Ws-*eds1* and *B. juncea* (*B.j*). Bands highlighted in orange indicate amplification of secondary inoculum on usually non-host plants upon primary inoculation with virulent *A. candida*. These experiments were repeated multiple times with similar results (see ***Supplementary file 6***).

effector from one race by introgression might offer an adaptive advantage assuming that the effector is not recognised.

Strikingly, each race shared a mosaic pattern of large genomic regions (>10,000 bp) that were virtually identical between two races (and diverged from the third). Given that mutations accumulate over time, this implies that the exchange must have occurred relatively recently. Yet, *A. candida* is an

obligate biotrophic parasite that is highly host-specific. The host-specificity of *A. candida* races was confirmed in our experiments, and this raises the question of how distinct *A. candida* races can achieve the physical contact required for them to have sex and recombine. We addressed this question using experimental infections of host plants with multiple races. Crucially, we showed that infection with a virulent race of *A. candida* suppresses host immunity sufficiently to enable subsequent co-colonization by an otherwise non-virulent race (see also *Cooper et al., 2008*). Although this opens up competition between races for the same (limited) resource, it also brings a significant evolutionary/ecological advantage by blurring the borders of host range, thereby creating secondary contact zones that enable sexual reproduction and genetic exchange between races. Given the high-host specificity of obligate biotrophic parasites, this ability of *A. candida* to enable occasional genetic exchange between specialised races could create novel repertoires of effector alleles that would enable colonization of new hosts ('host jumps'). For example, *Arabidopsis* resistance to race Ac2V can at least in part be explained by the *WRR4* gene, a classical NB-LRR *R* gene, which presumably recognizes an Ac2V effector (*Borhan et al., 2008*). Hypothetically, if this Ac2V effector would segregate away in hybrid offspring, or by loss of heterozygosity, these offspring could become virulent on *WRR4*-carrying *Arabidopsis* hosts.

Once a new hybrid race has been established, it can reproduce asexually and clonally on susceptible hosts without continued genetic exchange with other races. We base this inference on the exceptionally high genotypic similarity between independent isolates that infect the same host plant (i.e., AcBoT-AcBoL and AcEm2-AcNc2). For example, AcBoT and AcBoL share >97% of their heterozygous sites. This observation is significant because it rules out sexual reproduction (or selfing with recombination), given that Mendelian segregation would eradicate this high level of genotypic similarity within a single generation. Hence, we conclude that reproduction of AcBoT and AcBoL is asexual and clonal, and we speculate that these races may have derived from a common ancestor that was selected coincident with the onset of widespread *B. oleracea* cultivation in Europe. The *A. thaliana*-infecting isolates (AcNc2 and AcEm2) were sampled at a broad spatiotemporal scale (14 years and 100 miles apart), and yet they showed a very high nucleotide similarity (>99.95% identical). This shows that clonal reproduction coincided with their rapid population expansion. Remarkably though, compared to the former two isolates, AcEm2 and AcNc2 had a relatively low level of observed heterozygosity, and the number of heterozygous sites was >13 times less than that of AcBoT and AcBoL. Since these races do not self-fertilize, this suggests that gene conversion or another mechanism might be operating to reduce the nucleotide diversity within their genomes over time. Such loss of heterozygosity (LOH) has also been reported for *Phytophthora capsici* (*Lamour et al., 2012*), and in yeast, LOH events can encompass entire chromosomes, which is thought to be explained by the break-induced replication (BIR) mechanism (*Diogo et al., 2009*).

'Hybrid speciation' or 'recombinational speciation' is often associated with a mosaic genome (*Baack and Rieseberg, 2007*; *Stukenbrock et al., 2012*) and because introgressed genes have been 'pre-tested' by selection, they are more likely to be adaptive than random changes to the genetic code by mutations (*Hedrick, 2013*). For example, *Helianthus anomalus* is a wild sunflower species derived via hybridization between two parental species, and its genome is characterised by parental species blocks. In this case, however, hybridisation and speciation occurred over a short evolutionary time-span (10–60 generations) (*Ungerer et al., 1998*), which differs from hybridisation in *A. candida*, where it appears to be a persistent evolutionary process. Darwin's finches are a classic example of a young adaptive radiation, and recent research by *Lamichhaney et al. (2015)* showed that species from different islands show extensive genomic exchange through recent hybridisation. Introgressive hybridization was found to occur throughout the radiation, fuelling the genetic variation in beak shape, and thereby facilitating adaptive evolution. Perhaps more pertinently, the relationships of members of the parasite's principal host, plants of the *Brassica* genus, are themselves considered to be the result of a number of hybridisation events, as defined by the Triangle of U (*Nagahara, 1935*). Hybridisation events have also been implicated in speciation and adaptive radiation in vertebrates (*Nichols et al., 2015*). Furthermore, rare introgression events among *Heliconius* butterflies are believed to have facilitated the exchange of mimicry genes across multiple time points post speciation (*Martin et al., 2013*). Evidence for more frequent introgression has been observed in the malaria vector species complex (*Anopheles gambiae*) (*Fontaine et al., 2015*). More examples of such introgression can be anticipated to emerge given the continued improvement and reduction in

cost of DNA sequencing methods, and the development of novel software that enable analysis of whole genome recombination (Ward and van Oosterhout, submitted).

Perhaps more than any other system, studies on yeasts have offered valuable insights into how introgression and hybridisation can affect genome architectures of eukaryotic microorganisms (reviewed by *Dujon, 2010*). Genetic exchanges amongst three strains of *Saccharomyces cerevisiae* have been quantified using genomic data which show that the rate of outcrossing is remarkably low, with only 314 outcrossing events during circa 16 million cell divisions (*Ruderfer et al., 2006*). With such a low rate of genetic exchange, the strains accumulate sequence variation by mutation and thus genetically diverge. Given the low level of genetic exchange amongst the *A. candida* host races, this may also explain their genetic divergence. In yeast, natural hybrids have been reported in many species (*Dujon, 2010*), with hybridisation leading to the nonreciprocal genetic exchange accompanied by the loss of genes and genomic regions. This results in chimeric sequences from which novel lineages could emerge (*Greig et al., 2002*; *Usher and Bond, 2009*). For example, *Lachancea kluyveri* possesses mega-base long chromosomal fragments of distinct composition (*Payen et al., 2009*). Furthermore, the genomes of wine strains of *S. cerevisiae* contain introgressed regions from *Saccharomyces paradoxus*, *S. kudriavzevii kudriavzevii*, *S. uvarum uvarum*, and *Zygosaccharomyces bailii* (*Dujon, 2010*). Given that the introgressed sequences in the genome of *S. cerevisiae* are nearly identical to those in the donor genomes, hybridisation must have occurred recently, which is similar to what we observe in *A. candida*. Although introgression appears to be a general phenomenon in yeast genomes, its importance for evolution has yet to be determined (*Dujon, 2010*). The mosaic-like genome structure of *A. candida* suggests that hybridisation and genetic introgression may play an equally important role in the biology of this oomycete.

Introgression can introduce novel adaptive trait combinations (*Seehausen, 2004*; *Hedrick, 2013*) as well as enable the loss by segregation of host-specific 'avirulent' effector alleles that can trigger the immune response in potential hosts. In rare novel recombinant races, such maladapted effectors that trigger the response of a specific host may be segregated away, allowing the recombinant to avoid immune recognition and colonise this new host. Once this happens, the new hybrid can rapidly expand its geographic range and population size through clonal reproduction. Hybrids between *Phytophthora* spp. races can show expanded host range compared to their parental lineages (*Ersek et al., 1995*) while the importance of virulence gene transfer that subsequently leads to the expansion of pathogens' host range has also been reported for bacterial and fungal pathogens (*Doolittle, 1999*; *Mehrabi et al., 2011*).

The ability of pathogens to recombine and generate novel recombinant genotypes and subsequently proliferate clonally may be particularly favoured in the agro-ecological environment. Recent fusion between genetically distinct plant pathogens has been shown in *Mycosphaerella graminicola* (*Stukenbrock et al., 2012*), where a hybrid speciation event has generated a generalist pathogen of grass species, and in *Blumeria graminis* (*Hacquard et al., 2013*), where a mosaic genome structure has been generated by sex between divergent isolates. Adaptation of pathogens to agro-ecosystems can be correlated with a reduction in diversity of recently emerged lineages and at the same time, high levels of genome plasticity (*Stukenbrock and Bataillon, 2012*). This genome plasticity has also been observed in *B. graminis* (*Hacquard et al., 2013*). Race specialization observed in the potato late blight pathogen *P. infestans*, the rice blast pathogen *Magnaporthe oryzae*, and the wheat yellow rust pathogen *Puccinia striiformis* (*Stukenbrock and Bataillon, 2012*), may represent the result of a broader mechanism of pathogen adaptation to a crop monoculture. *A. candida*, which is known to live on both wild weeds (e.g., *A. thaliana*) and important crop species (e.g., *B. juncea*), thus provides remarkable insights into the impact of recombination in generating new virulent races and subsequent clonal propagation of a novel race. These findings are particularly relevant to modern agricultural methods and the emergence of new epidemic pathogen strains on crop monocultures.

## Materials and methods

### Pathogen isolation and cultivation

*A. candida* races were isolated and propagated by first washing zoosporangia from infected leaves and then infecting *A. thaliana* Ws-*eds1* (enhanced disease susceptibility [*Parker et al., 1996*]) plants. After 2 weeks, one pustule was punched out and spores were treated on ice for 30 min to release zoospores. Unhatched zoosporangia were removed by filtration and zoospores

were diluted to ~10 zoospore per ml and sprayed on *A. thaliana* Ws–*eds1* plants (~100 µl/plant). This procedure was repeated four times until spores were bulked up on *A. thaliana* Ws-*eds1* plants. Zoosporangia were harvested using a homemade cyclone spore collector (Mehta and Zadoks, 1971). Subsequently, *A. candida* races AcEm2/AcNc2, AcBoT/AcBoL and Ac2v were propagated and maintained on *A. thaliana* Ws-2, *B. oleracea* and *B. juncea*, respectively.

## Virulence test

Host specificity was tested for the AcNc2, Ac2V and AcBoT races on a number of *A. thaliana* and *Brassica* spp. accessions (*Supplementary files 1, 2*). *A. candida* inoculations were performed using the following method: zoospores were suspended in water ($10^5$ spores/ml) and incubated on ice for 30 min; the spore suspension was then sprayed on plants using a spray gun (~700 µl/plant), and plants were incubated in a cold room in the dark over night. Infected plants were kept under 10-hr light and 14-hr dark cycles with 20°C day and 16°C night temperature. Plants were scored susceptible if a pathogen was capable of accomplishing its life cycle and sporulation was macroscopically visible within 3 weeks after plant inoculations.

For sequential infection analyses, we developed *A. candida* race specific PCR primers from genome sequences (*Supplementary file 5*). Regions in all vs all alignments were identified that lacked read coverage by other isolates. Primers were designed within these regions to amplify products of between 300 and 800 bases and were tested on pure genomic DNA extracts from each isolate.

Primary inoculum was sprayed onto control and test plants. In the case of AcNc2 defence suppression assays, both *A. thaliana* Ws-2 and Ws-*eds1* were inoculated; for AcBoT assays, *B. oleracea* and Ws-*eds1* were inoculated; for Ac2v assays, *B. juncea* and Ws-*eds1* were inoculated. Following inoculations, plants were incubated in the dark in a cold room over night before transferring to a growth cabinet set to the conditions described above. In addition to pathogen treatment, the same numbers of plants were also treated with water in order to serve as a negative infection control. At 7 days post-inoculation a secondary infection with the avirulent *A. candida* race was performed on 50% of the plants while the remaining 50% were water treated (*Supplementary file 6*). Plants were returned to the growth cabinet and cultivated for a further 8 days. Inoculated plant tissue was harvested, washed in sterile water to remove surface adhering spores, and flash frozen in liquid nitrogen. DNA was prepared using a DNeasy Plant Mini Kit (Qiagen, Valencia, CA) as described in manufacturers instructions. PCR was performed using race-specific primers and products visualised on a 1% agarose gel.

Plants from which tissue was harvested were maintained for a further 7 days before re-inoculation onto the original host of the otherwise non-virulent pathogen. This was done in order to confirm the completion of the secondarily inoculated pathogen race's lifecycle on immunosuppressed non-host plants.

## DNA extraction and sequencing

DNA was extracted from zoosporangia according to the method described in *Mckinney et al. (1995)*, and Illumina libraries for sequencing were constructed according to *Farrer et al. (2009)*. Paired-end libraries of 800 bp and 400 bp insert lengths (for the race Ac2V only one library of ~400 bp) were sequenced using Illumina Genome Analyzer II platform at the Sainsbury Laboratory Sequencing Centre (GA2). The base calling was done on the Illumina GAP v1.3 pipeline.

## cDNA extraction and sequencing

*A. thaliana* Ws-0 plants were infected with *A. candida* AcNc2 and infected plants were harvested at 0, 2, 4, 6, 8 and 10 days after infection. RNA was isolated using TRI Reagent RNA Isolation Reagent (Sigma, UK), and subsequently enriched for mRNA with Dynabeads (Invitrogen). cDNA was prepared using the SMART cDNA Library Construction Kit (Clontech, Sunnyvale, CA) according to manufacturer's instructions. These libraries were normalized using Evrogen Duplex-specific nuclease (DSN). Normalized cDNA libraries were fragmented using Covaris sonicator and libraries prepared according to Illumina genomic library preparation kit. Libraries were sequenced on the Illumina GA2 platform.

The sequence data have been deposited at the EMBL Nucleotide Sequence Database, with the accession numbers for *A. candida* AcNc2: SRR1811450, SRR1811464, AcEm2: SRR1806791, AcBoT: SRR1811472, SRR1811473, AcBoL: SRR1811474, Ac2V: SRR1811471.

## Genome and transcriptome assembly

The genomic assemblies were produced using program Velvet 1.0.19 (*Zerbino and Birney, 2008*). Using BLAST (*Altschul et al., 1990*), resulting contigs were searched (BLASTN, E-value ≤ $10^{-5}$) against genomic sequences of *A. thaliana* TAIR 9.0 (*The Arabidopsis Genome Initiative, 2000*), fungi *Neurospora crassa* (*Galagan et al., 2003*), a collection of bacterial genomes (such as, *Xanthomonas* sp. and *Pseudomonas* sp.: microbialgenomics.energy.gov), and against mitochondrion of *Pythium ultimum* (*Levesque et al., 2010*) to remove potential contamination and mitochondrial DNA. The assembly of AcNc2 was processed by merging the overlapping contigs from two velvet assemblies (based on the k-mers 55 and 61) with the Minimus2 genome merge pipeline (*Sommer et al., 2007*) and in-house perl scripts.

Read alignment and mapping was performed using programs BWA (0.7.3) and SAMtools (0.1.17) (*Li and Durbin, 2009*; *Li et al., 2009*) and BedTools (*Quinlan and Hall, 2010*). Duplicates were removed from mapped reads and SNP calling and filtering was done with BCFtools view and varFilter (-D100). Where required, conversion to fasta format from vcf was done using the modified version of vcf2fq which has been modified to include indels (http://sourceforge.net/p/vcftools/feature-requests/19/) and then sequences were aligned using mafft online server (http://mafft.cbrc.jp/alignment/server/).

Illumina sequenced cDNA from the AcNc2 infected *A. thaliana* Ws-0 leaves was assembled using Velvet/Oases (*Schulz et al., 2012*) with different k-mer lengths (43, 45, 47, 51, 55, 57, 61, 63). We used various characteristics (total number of contigs, assembly size, longest contig length and mean contig length, and the proportion of core eukaryotic genes (KOGs) predicted by CEGMA) to assess assembly quality. Two best assemblies based on the k-mers 55 and 57 were merged using VMATCH (http://www.vmatch.de/). The cDNA orientation was predicted using Illumina generated cDNA 5′ tags. Using Bowtie aligner (*Langmead et al., 2009*), cDNA 5′ tags were aligned against the assembled cDNA and, based on tag counts, orientated in the 5′–3′ direction.

## Gene prediction and annotation

*Ab initio* gene predictions were performed for *A. candida* AcNc2 using the Augustus gene prediction package (*Stanke et al., 2006*), Geneid (*Blanco et al., 2002*) and GeneMark (*Lomsadze et al., 2005*). Alternative splice variants were predicted with Augustus. To improve gene predictions, the 'hints' files were created using cDNA evidence and gene homology information. The generated library of AcNc2 transcripts was aligned to the AcNc2 assembly using BLAT (*Kent, 2002*), setting minimal identity to 92; the 'hints' file was produced with script blat2hints.pl provided with the Augustus package. The parameters previously obtained for the gene prediction in the *A. laibachii* genome project were utilized when running the Augustus and Geneid. GeneMark predictions were made with the default settings. Consensus gene models were generated with Evigan (*Liu et al., 2008*). Subsequently, the catalog of non-overlapping gene models was created from the Evigan, Augustus, Genemark and Geneid predictions. The gene space coverage was assessed with CEGMA (*Parra et al., 2007*).

Functional annotations of AcNc2 proteins were performed via comparison of the predicted protein sequences with the protein databases; UniProtKB (*Suzek et al., 2007*) and NCBI non-redundant RefSeq (*Pruitt et al., 2009*) databases were scanned using BLASTP algorithm (E-value ≤ $10^{-5}$); and Pfam database (*Punta et al., 2012*) was searched with the program hmmscan from HMMER3 (*Eddy, 2011*) with the default settings. GO terms were assigned with BLAST2GO pipeline (*Conesa et al., 2005*).

Gene families were predicted using Tribe-MCL algorithm that implements Markov cluster (MCL) approach for the clustering of proteins into families based on the pre-computed sequence similarity information (Enright et al., 2002).

Signal peptides and cleavage sites were predicted by the hidden Markov Model and the neutral network algorithm implemented in SignlP 4.0 program (*Petersen et al., 2011*). Transmembrane helices were predicted with the hidden Markov Model in TMHMM 2.0 (*Moller et al., 2001*).

Candidate cytoplasmic effectors carrying 'RXLR' motif were identified through the string search of the predicted secreted proteins using 'R[A-Z]L[RQ]' regular expression in the first 100 residues downstream of the signal peptide cleavage site. Crinkler's homologs were detected using BLASTP searches (E-value ≤ $10^{-5}$) against NCBI non-redundant RefSeq database (*Pruitt et al., 2009*). Homologs of *A. laibachii* 'CHXC' proteins were identified through Tribe-MCL clustering, also using

BLASTP searches (E-value ≤ $10^{-5}$) of the *A. laibachii* predicted proteome, and string search for the 'CH[A-Z]C' motif in the first 100 residues after predicted signal peptide.

## Repetitive elements

The library of repeats in AcNc2 assembly was constructed with RepeatScout (*Price et al., 2005*) and was joined with the previously made library for *A. laibachii* (Kemen et al., 2011). This updated library in combination with RepeatMasker (http://www.repeatmasker.org/) was used for the identification of repeats and their frequencies. Transposon elements were annotated using TBLASTX searches (E-value ≤ $10^{-5}$) against the database of transposon elements, RepBase (*Jurka et al., 2005*).

## Phylogenetic analysis

Phylogenetic trees were built (MrBayes program [*Huelsenbeck and Ronquist, 2001*]) for the nucleotide sequence alignments of the orthologous genes present in four *A. candida* races and *A. laibachii*. Trees were built for 100 single copy genes (50 core eukaryotic genes and 50 singletons with unknown function) Phylogenetic Bayesian inference and Markov chain Monte Carlo (MCMC) methods were used to estimate the posterior distribution of model parameters. We used lset = 6, gamma model, mcmc of 10 million, samfreq = 7000 and burnin = 375. Population genetic parameter 'theta' ($\Theta = 4N_e\mu$) was estimated using mlRho program (*Haubold et al., 2010*). Four different topologies were inferred with equal support which warranted further investigation into the role of introgression.

Bayesian Evolutionary Analysis by Sampling Trees (BEAST) software package version 1.7 (*Drummond and Rambaut, 2007*) was used to produce the race phylogeny (based on contig 1). BEAST implements Markov chain Monte Carlo (MCMC) algorithms for Bayesian for divergence time dating (*Drummond and Rambaut, 2007*). Bayesian phylogenetic trees were constructed with a HKY+G nucleotide substitution model under a strict molecular clock (with units in mutations per site) and a Yule tree prior. We ran ten independent MCMC analyses each of 10 million steps and a 10% burn-in. MCMC chain mixing was assessed using Tracer 1.5 which showed ESS >3000 for each statistic.

## Recombination analyses and detection of exchanged sequence blocks

Recombination events were statistically identified on contigs ≥10,000 bp using the software RDP3 using five independent detection algorithms: RDP (*Heath et al., 2006*), GENECONV (*Padidam et al., 1999*), Maxchi (*Smith, 1992*), Chimaera (*Posada and Crandall, 2001*), and 3Seq (*Boni et al., 2007*). Tests were conducted using a critical value α = 0.05 and p-values were Bonferroni corrected for multiple comparisons of sequences. Sequences were linear using unphase base calling and the random assignment of one of the nucleotides at each polymorphic site. Given that recombination algorithms use *cis* mutations to define regions of a sequence that share the same phylogenetic history, the statistical power to detect recombination is reduced when using unphased data (Darren Martin pers. comm.). This is because the signal of unphase base calling erodes any underlying signal of recombination. This procedure is conservative and underestimates the true number of recombination events because it reduces sequence similarity between the recombinant and parental sequence. Phylogenetic evidence of recombination was required to confirm a recombination event. Window sizes for each detection method were set to defaults. Only events for which the software identified the parental sequences (i.e., no 'unknowns') without ambiguous start and end position of the recombination block are reported and used in the analyses. Furthermore, events were only considered to be genuine if they were supported by at least three of the five detection algorithms. Hence, the estimates of the number of recombination events are conservative.

The effects of recombination on the sequence similarity between three genomes was visualised using a newly developed code in the R package HybRIDS (Hybrid Recombination, Identification and Dating, Software, http://www.elsa.ac.uk/). HybRIDS uses a colour triangle to visualise the sequence similarity between aligned sequences. It calculates the colour of each 100 bp window based on the proportion of SNPs shared between the pairwise sequences. All monomorphic sites were excluded in this calculation. HybRIDS uses the additive colour system in which the primary colours used are red, green, and blue. These colours are plotted on the corners of the RGB colour triangle, which is shown in the legend as a reference. In cases where all SNPs are shared between just two of the three races, the hybrid colour is an exact 50% mix of two primary colours. The hybrid colours are yellow, purple and turquoise, and these colours suggest recent gene exchange between the two races. At such

recombined regions, the third race receives its primary colour (because by definition, it must be unique at the 100 bp window and completely dissimilar from the other two races). Older recombined regions are predicted to have accumulated SNPs unique to each race, causing the block of two hybridizing races to diverge. In the graph, the colour of such areas contains more than 50% of the primary colour of that race. The colours in the centre of the triangle are pale and reflect areas where the three races share approximately similar numbers of polymorphisms. The SNPs in these pale regions, and in regions where the colours are close to the primary colours, are more likely caused by mutations than by genetic exchange. Note that primary colours can also be assigned to regions that may have recombined, but which originate from a race that was not sampled, and hence, which polymorphisms were not included in this analysis.

## Dating recombination events

We used the sequence divergence of the recombination block between the recombinant and the minor parent (i.e., the sequence donating the recombinant region) to estimate the divergence time since a recombination event. We used two dating methods, a binomial mass function and an analysis with the Bayesian Evolutionary Analysis by Sampling Trees (BEAST) software package version 1.7 (*Drummond and Rambaut, 2007*). A binomial mass function was used to estimate the mean divergence time of a block of given size with an observed number of SNPs. In order to correct for mutation saturation, homoplasy, back mutations and transition/transversion ratios, we converted the observed number of SNPs into the number of mutations using a JC correction (*Jukes and Cantor, 1969*). The probability of finding a number of SNPs less or equal to the observed number in a block of known size was calculated. The mean time is found when the binomial mass function returns a probability value p = 0.5. This approach finds the most probable age of the recombination event, and it assumes that since the recombination event, the block evolved neutrally over t years, and that each base has the chance to mutate with a probability μ per year ($\mu = 10^{-6}$ and $10^{-7}$). The algorithm uses a strict molecular clock, and because the mutation rate in oomycetes is unknown, we assumed $\mu = 10^{-6}$ as well as $\mu = 10^{-7}$ per base per year. The lower value of the mutation rate of $\mu = 10^{-7}$ was used as a more conservative estimate. Given that we do not know the mutation rate of oomycetes, the estimated dates are merely an approximation and shown to illustrate that the exchanged blocks are dated back to a wide range of evolutionary times. The simple dating method based on the binomial mass function was compared to more computationally intensive analysis with BEAST by dating 20 recombination events from the 'contig 1' of AcNc2 and performing a linear regression analysis to confirm application of the faster binomial algorithm to all 675 recombination events. The principle aim of these analyses was to identify whether or not recombinant regions span a range of dates (in line with the expectation under an introgression model).

BEAST bayesian phylogenetic trees were constructed with a HKY+G nucleotide substitution model under a strict molecular clock ($\mu = 10^{-6}$) and a Yule tree prior. We ran 10 independent MCMC analyses each of 10 million steps and a 10% burn-in for each of the 20 recombination events. MCMC chain mixing was assessed using Tracer 1.5 which showed ESS >3000 for each statistic.

Note however that the divergence estimates made by the binomial mass function and the analysis with BEAST are conservative (i.e., the true time of the recombination event are probably more recent) given that we had only three sequences in the analysis. Consequently, the 'true' parental sequence has probably not been sampled, which means that the observed divergence is larger than that of the actual parental (donor) sequence.

## Substitution rates and selection

The substitution rates (non-synonymous substitution rate per non- synonymous site (dN) and synonymous substitution rate per synonymous site (dS)) and the ratio of the dN/dS for the orthologous protein-coding sequences between three isolates were estimated with the M0 model in PAML (Yang, 2007). The dN/dS ratio is traditionally used as an indicator of the strength and type of selective constrains acting upon a gene. Values of dN/dS ≈ 1 indicate neutral evolution. Values of dN/dS significantly less than unity indicate purifying selection, whereas dN/dS significantly larger than unity suggest positive selection. The Tajima's *D* statistics (*Tajima, 1989*) were not calculated because we do not possess allele frequency data.

## Acknowledgements

We thank Dr M Borhan for the providing Ac2V material; Jodie Pike for technical assistance; Dr OJ Furzer, M Burrell, Dr C Schudoma and Dr D MacLean for computational assistance. We thank Sophien Kamoun and Kentaro Yoshida for helpful comments on earlier drafts of the manuscript. The BEAST 1.6.2 analyses were carried out on the High Performance Computing Cluster supported by the Research and Specialist Computing Support service at the University of East Anglia. TSL was supported by the grant DASTI (Danish Agency for Science, Technology and Innovation) and TSL, MM, AG, KB, EK, VC, AR-S and AB on the European Research Council Advanced Investigator grant 233376 (ALBUGON); CvO, MM and BW are funded by Earth & Life Systems Alliance (ELSA). EBH was supported by the grant NE/H020691 from NERC (UK Natural Environment Research Council).

## Additional information

### Funding

| Funder | Grant reference | Author |
|---|---|---|
| European Research Council (ERC) | 233376 (ALBUGON) | Jonathan DG Jones |
| Norwich Research Park | Earth & Life Systems Alliance (ELSA) | Mark McMullan, Ben J Ward, Cock van Oosterhout |
| Natural Environment Research Council (NERC) | NE/H020691 | Eric Holub |
| Uddannelses- og Forskningsministeriet | Danish Agency for Science, Technology and Innovation | Jonathan DG Jones |

The funders had no role in study design, data collection and interpretation, or the decision to submit the work for publication.

### Author contributions

MM, Conducted bioinformatics analysis, Wrote the manuscript, Conducted analysis of recombination and performed simulations, Conception and design, Acquisition of data, Analysis and interpretation of data, Drafting or revising the article; AG, Conducted bioinformatics analysis, Wrote the manuscript, Conception and design, Acquisition of data, Analysis and interpretation of data, Drafting or revising the article; KB, Performed the co-infection assays and molecular experiment and analysis, Acquisition of data, Analysis and interpretation of data; EK, Isolated races and established propagation protocols, Performed virulence test, Acquisition of data; BJW, Conducted analysis of recombination and performed simulations, Analysis and interpretation of data; VC, Performed the co-infection assays and molecular experiment and analysis, Performed virulence tests, Acquisition of data, Analysis and interpretation of data; AR-S, TS-L, Performed virulence tests, Acquisition of data; AB, Acquisition of data, Analysis and interpretation of data, Contributed unpublished essential data or reagents; EH, Acquisition of data, Drafting or revising the article, Contributed unpublished essential data or reagents; CO, Wrote the manuscript, Conducted analysis of recombination and performed simulations, Designed the experiments and corrected the manuscript, Conception and design, Analysis and interpretation of data, Drafting or revising the article; JDGJ, Conception and design, Acquisition of data, Analysis and interpretation of data, Drafting or revising the article

### Author ORCIDs

Torsten Schultz-Larsen, http://orcid.org/0000-0001-6759-0074

## Additional files

### Supplementary files

• Supplementary file 1. List of *Arabidopsis thaliana* and *Brassica* spp. accessions assayed in virulence tests with *Albugo candida* race type isolates AcBoT, Ac2V and AcNc2.

• Supplementary file 2. List of *Arabidopsis thaliana* accessions assayed in virulence tests with *Albugo candida* isolates AcNc2 and Ac2v.

• Supplementary file 3. Polymorphisms between *Albugo candida* race genomic regions verified with Sanger sequencing.

• Supplementary file 4. List of the predicted recombination blocks generated by the RDP3 software.

• Supplementary file 5. Details of race-specific primers.

• Supplementary file 6. Details of co-infections assays.

### Major datasets

The following datasets were generated:

| Author(s) | Year | Dataset title | Dataset ID and/or URL | Database, license, and accessibility information |
|---|---|---|---|---|
| The Sainsbury Laboratory | 2015 | Genomic sequencing of Albugo Candida Ac Nc2 | http://www.ncbi.nlm.nih.gov/sra/?term=SRR1811450 | Publicly available at NCBI Short Read Archive (SRR1811450). |
| The Sainsbury Laboratory | 2015 | Genomic sequencing of Albugo Candida Ac Nc2 | http://www.ncbi.nlm.nih.gov/sra/?term=SRR1811464 | Publicly available at NCBI Short Read Archive (SRR1811464). |
| The Sainsbury Laboratory | 2015 | Genomic sequencing of Albugo Candida Ac BoT | http://www.ncbi.nlm.nih.gov/sra/?term=SRR1811473 | Publicly available at NCBI Short Read Archive (SRR1811473). |
| The Sainsbury Laboratory | 2015 | Genomic sequencing of Albugo Candida Ac BoT | http://www.ncbi.nlm.nih.gov/sra/?term=SRR1811472 | Publicly available at NCBI Short Read Archive (SRR1811472). |
| The Sainsbury Laboratory | 2015 | Genomic sequencing of Albugo Candida Ac BoL | http://www.ncbi.nlm.nih.gov/sra/?term=SRR1811474 | Publicly available at NCBI Short Read Archive (SRR1811474). |
| The Sainsbury Laboratory | 2015 | Genomic sequencing of Albugo Candida Ac 2v | http://www.ncbi.nlm.nih.gov/sra/?term=SRR1811471 | Publicly available at NCBI Short Read Archive (SRR1811471). |
| The Sainsbury Laboratory | 2015 | Genomic sequencing of Albugo Candida Ac EM2 | http://www.ncbi.nlm.nih.gov/sra/?term=SRR1806791 | Publicly available at NCBI Short Read Archive (SRR1806791). |

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
