## [Decision Letter]

[Editors’ note: this article was originally rejected after discussions between the reviewers, but the authors were invited to resubmit after an appeal against the decision.]

Thank you for choosing to send your work entitled “Immunosuppression enables expanded host ranges and can explain mosaic genome structures in *Albugo candida* races” for consideration at *eLife*. Your full submission has been evaluated by Detlef Weigel (Senior editor and Reviewing editor) and two peer reviewers, and the decision was reached after discussions between the reviewers. Based on our discussions and the individual reviews below, we regret to inform you that your work will not be considered further for publication in *eLife*.

First, there were concerns about the evidence for hybridization, both from the side of in vivo experimentation and from the interpretation of the sequence data with respect to ploidy and recombination patterns. Second, the reviewers felt that the failure to identify credible candidate genes or gene networks that arose from hybridization and that are responsible for adaptation weakened the impact of the work.

Reviewer #1:

The article by McMullan and colleagues investigates the genome of the obligate pathogen *Albugo candida* infecting Brassicas including Arabidopsis. The authors sequenced the genomes of 5 strains sampled from different Brassica hosts falling into 3 distinct clades. Analysis was based mostly on 3 strains, one from each race (clade). Authors identified recombination blocks and regions of divergence among the three races. The authors combine genome sequencing, modeling and genetic work to assemble a nice story on divergence among three races of the pathogen.

The authors should formally test for linkage disequilibrium and clonality using SNPs and the index of association or LD.

I liked the inclusion of simupop simulation to infer patterns and date events. This effort showed that incomplete lineage sorting does not lead to the observed mosaic genome structure. However, I believe Figure 4 of the supplements is missing and I could not see the actual graphs showing this pattern.

The phenotyping work on host range was not repeated in independent experiments and the experiments included a different number of treatments for each strain.

Overall, the manuscript as presented lacks novelty although the work is well done and highly publishable elsewhere. The genome of *A. candida* was published in 2011 by Links et al. Here the authors resequence 5 strains and analyze 3 strains in fine detail. The races are found to have a mosaic like genome structure and that races diverged a considerable amount of time ago. *A. candida* is host specific, and thus races associated with a given host are not expected to cross breed with those on another host; the authors established presence of recombination blocks. They also demonstrate that co-inoculated strains from a host and non-host on a host can establish infection by the non-host strain and replicate this effect for several strain-host combination (Figure 6). However, this work is not repeated and lacks the validation from a full second experiment. Furthermore, it only shows the strains can coexist in the same environment, but it does not demonstrate hybridization or recombination. I think if they could demonstrate sexual reproduction or hybridization among diverged races, and tie the hybrid genomes into this story they would have a great paper. As it stands, I think this would be better published in a disciplinary journal given that it mostly provides information on more distant divergence, adaptation and coinfection. Lack of sex also needs to be more formally tested on a larger sample of strains.

Reviewer #2:

This manuscript by McMullan et al describes patterns of genomic variation and exchange between three races of *Albugo candida*, an oomycete pathogen of Brassicas, and provides evidence that co-infection could allow for such genetic exchange. Virulence tests confirmed prior work on the host-specificity of these isolates (AcNc2, Ac2V and AcBot) *to A. thaliana*, *B. juncea* and *B. oleracea*, respectively. When host plants are first infected by the specific pathogen race, this can allow for subsequent infection of otherwise avirulent races in the same plant, expanding on prior work showing secondary infection by downy mildews on hosts that have primary resistance. By analyzing genome assemblies of five isolates, the authors identified regions that appear to have ancestrally recombined between the races, resulting in a “mosaic genome”, followed by an apparent clonal expansion. The authors hypothesize that immunosuppression by the infectious isolates allowed a non-infectious race to colonize and recombine with it, allowing for new combinations of virulence genes.

This is an interesting study of genetic variability and exchange between pathogenic lineages; however the methods used to detect recombination do not clearly establish the certainty of inferred exchange events. The risk of relying on a single program (RDP3) to determine recombination is highlighted by the primary publication on this method, which states: “The drawback of such a flexible, exploratory framework is that it can often be difficult to assess the uncertainty associated with inferred recombination patterns. However, with its wide range of cross-checking tools, RDP3 is complementary to probabilistic recombination analysis approaches.” The authors have not included any such support for the recombinogenic regions identified using this data, including standard metrics of linkage disequilibrium, and it is unclear if their measure of using events detected by 3 of 5 models in RPD3 represents an optimal concordance. Some simulations are included, although this is used to evaluate the origin of the overall pattern, rather than validate the genomic regions detected with RDP3.

One major point of confusion in reading the manuscript is the lack of clarity on the ploidy level of *A. candida* and the methods used to analyze nucleotide variation. The authors compared heterozygous sites (suggesting the species is diploid), although there is no description in the supplementary methods of how these sites were identified or their confidence level. The “unphase base calling and the random assignment of one of the nucleotides at each polymorphic site” provided the sequence used for the recombination analysis. This process is then described as “conservative”, based on unphased heterozygous data underestimating variance, but direct support for this statement is not provided. The methods and the analysis of how nucleotide variants were identified through alignments of raw sequence or assemblies need to be more thoroughly described.

Given that ancestral recombination between these races is the main result of this paper to support their hypothesis, this analysis needs to be supported by an additional method, and a more detailed description of how results compare from the individual methods in RDP3 and probability based methods.

[Editors’ note: what now follows is the decision letter after the authors submitted for further consideration.]

We have considered your appeal regarding your manuscript entitled “Immunosuppression enables expanded host ranges and can explain mosaic genome structures in *Albugo candida* races”. We have now received comments from an expert on hybridization, and in light of the very positive evaluation, we would like to invite you to revise your article.

Your manuscript presents the analysis of several genomes of different races of *Albugo candida*. You provide very nice evidence that the genomes show signs of introgression of up to 25% material from other races. Since these races have host ranges which, to the best of the authors' knowledge, exclude each other, it is of great interest to understand how recombination between races can occur. You solve this conundrum by showing that exposure by one pathogen race makes the host plant susceptible to other races, to which they are normally resistant, thus providing a clear path to hybridization between races that normally have nonoverlapping host ranges.

The study is well carried out and convincing, but the presentation could be improved in several places.

1) While the material provided is clearly suitable for *eLife*, the manuscript is not written in a very accessible style, rather what one would expect for a much more specialist journal. The message of this study is of potential interest for many readers (not only plant pathologists), but there is the danger it will attract little attention as currently presented. For example, proper population genetic terminology is mostly missing, examples from other system are almost absent and comparisons with other cases of hybrid formation (there are many in plants and animals and several nice genomic studies have been published) are ignored. The topic of breakdown of isolation mechanisms has implications for other central topics in evolution and ecology, such as local adaptation and speciation. Here the discussion could go much further.

2) The data suggest that 25% of the genome of a race is introgressed from a genome from a different host race. With this much introgression, and assuming that this study did not find exceptional cases, but rather a representative sample from the wild, one wonders about the genetic architecture of host-range determination. If host-ranges are determined by multiple loci, one would except that recombination will quickly lead to super genotypes, with very wide host ranges. Apparently this did not happen; maybe recombination cannot change this. This would be the case if they are sitting in non-recombining gene clusters, or a single locus (with multiple alleles) is responsible for host ranges. We would like to see some discussion about this.

3) The authors use the term “immunosuppression”. To an animal immunologist, this implies that the study included the assessment of immunological parameters, which is not the case. Rather, you observe in experimental infections that a first infection facilitates infection with a second race of the pathogen. We suggest rephrasing this, so as to not confuse animal immunologists.

---

## [Author Response]

*[Editors’ note: the author responses to the first round of peer review follow*.*]*

Regarding the statement that “the reviewers felt that the failure to identify credible candidate genes or gene networks that arose from hybridization and that are responsible for adaptation weakened the impact of the work,” there is nothing about this in the reviewers’ substantive comments, and so we do not understand how this can be used to justify rejection. This comment is asking us to solve the molecular basis of non-host resistance in non-colonized plants to these different races. Researchers spend their careers trying to understand non-host resistance, and it is unrealistic to expect a meaningful and decisive explanation to emerge from comparison of 5 isolates of 3 races that infect three different host plants. We have of course generated a list of candidate genes that might underpin non-host resistance in Arabidopsis to the Brassica races, but we want to test them to see which of these candidate genes are causal, and also characterize the corresponding resistance genes that underpin non-host resistance. We could certainly say “it might be one of these 10 polymorphic secreted genes that underpins host specificity”, but it would be better to combine that with empirical tests in a different paper.

We can easily add more detail about our mapping pipeline to the methods. We failed to spell out that the co-inoculation experiments were done 3 times (AcBoT followed by Ac2v on *B. oleracea*, Ac2v followed by Nc2 on *B. juncea* three times) or twice (Nc2 followed by Ac2v on Ws-2; another rep easily added if required).

In addition, the literature is very clear on the host-range specificity of each *A. candida* race, but given our narrative, we decided to try to go the extra mile by testing whether any of 100s of Arabidopsis accessions *might* be susceptible to the Brassica-infecting races. We found none. We inoculated trays with 24 modules, with each module containing 10-20 plants for each genotype and each tray contained 5 out of 24 modules that carried susceptible control plants: four at the edges and one in the middle so we can be sure that spores are distributed well. With these susceptible positive controls to verify functional inoculum, and scores of plants tested from each accession, these confirmatory data are decisive. If any Arabidopsis genotype had been susceptible to Brassica-infecting races, we would have seen it, and we are confident there is no risk of false negative conclusions.

We show categorical and unambiguous evidence for recombination between races in the mosaic genomes. Software RDP3 conducts 5 different tests of this, with 5 different algorithms (one of which is RDP). The reviewer noted that “… RDP3 is complementary to probabilistic recombination analysis approaches.” We designed new software (HybRIDS) which does include a probabilistic recombination analysis (see http://www.norwichresearchpark.com/HybRIDS). In a previous version of the manuscript we explained in detail how this probabilistic recombination analysis works and presented the results. However, we decided to cut this out to avoid redundancy and because we believe the evidence based on the various algorithms in RDP3 is normally accepted to be robust (especially given the fact we identified 675 significant recombinant blocks). However, we would be happy to include a table with exact probabilities of all 675 events. Also, we can easily include a (standard) LD analysis that will confirm that the observed mosaic genome structure is consistent with recombination. To be honest, this is superfluous to the much more sophisticated battery of analyses we have done already which shows beyond any doubt that the mosaic genome can *only* be explained by recent recombination between significantly diverged races.

This same reviewer is confused about ploidy. The oomycete plant pathogens such as *Phytophthora infestans, Hyaloperonospora arabidopsidis* and *Albugo candida* are all diploids, and occasionally of higher ploidy; the point is they are not haploid. We regret any lack of clarity in our manuscript that might have led to confusion about ploidy on the part of one reviewer.

The only weakness in the manuscript that we are prepared to acknowledge is that although we showed (in multiple reps) that immunosuppression enables coexistence of races that do not have overlapping host range, we were nevertheless (despite considerable effort) unable to detect formation of new hybrids between races (we looked for oospores in leaves co-infected by different non-selfing races). But *A. candida* is usually heterothallic; the races we tested could all be the same mating type (and nothing is known about genes that specify mating type in this species), which would make crosses impossible. The reviewers did not appreciate that we simply wanted to show the in planta phenomenon (coexistence due to immunosuppression resulting in increased host susceptibility). Because we failed to observe heterozygotes, reviewers saw this as a negative result. However, our analysis suggests that the rate of (genetic exchange + recombination) is 100-1000 times lower than the mutation rate (shown in further population genetic simulations not presented in the paper, but easily provided for supplementary information), so one would expect genetic exchange to be rare.

Reviewer 1 says: “Overall, the manuscript as presented lacks novelty although the work is well done and highly publishable elsewhere.”

We contest this assessment of novelty. The significance of our work is that it shows that races that colonize different Brassicaceae, and which were previously regarded as being incapable of genetic exchange with each other, can in fact coexist due to immunosuppression, and thus (if of opposite mating type) can mate. This observation, combined with the unambiguous evidence for mosaic genomes, provides a truly novel insight into how parasites can coevolve with their hosts, enabling substantial diversification while retaining potential for further release of novelty by occasional inter-race mating and recombination. Despite their considerable genetic diversification (∼1%, which is equivalent to the divergence between humans and chimps), *A. candida* races exchange so much genetic material that ∼25% of their genome is from another race. We show this beyond any doubt (using five independent algorithms as well as a probabilistic recombinant analysis in HybRIDS software) and use computer simulations to exclude other evolutionary scenarios. This level of genetic exchange + recombination between specialised pathogen races has *never* been observed in the wild at this scale. But most novel of all, we believe, are the evolutionary and ecological implications; we show that very rarely, the new recombinants are extremely fit and colonise a given host plant over a large geographic range through clonal population expansion. We demonstrate this using population genetic simulations that show that clonal population expansion is the only plausible origin of the AcBoT and AcBoL (*B. oleracea* infecting) race as well as the AcNc2/AcEm2 race. We show that for AcBoT and AcBoL, this must have happened recently (probably in the last few hundred years, probably when *Brassica oleracea* began to be widely grown in Europe), which has profound implications for agriculture.

*[Editors’ note: the author responses to the re-review follow*.*]*

*We have considered your appeal regarding your manuscript entitled “Immunosuppression enables expanded host ranges and can explain mosaic genome structures in* Albugo candida *races”. We have now received comments from an expert on hybridization, and in light of the very positive evaluation, we would like to invite you to revise your article*. *[…]*

*1) While the material provided is clearly suitable for* eLife*, the manuscript is not written in a very accessible style, rather what one would expect for a much more specialist journal. The message of this study is of potential interest for many readers (not only plant pathologists), but there is the danger it will attract little attention as currently presented. For example, proper population genetic terminology is mostly missing, examples from other system are almost absent and comparisons with other cases of hybrid formation (there are many in plants and animals and several nice genomic studies have been published) are ignored. The topic of breakdown of isolation mechanisms has implications for other central topics in evolution and ecology, such as local adaptation and speciation. Here the discussion could go much further*.

Many thanks for these suggestions to broaden the appeal and significance of the manuscript. We have gone to some length to broaden the appeal of our findings by considering them within a broader context of other pathogens, yeast, plants, malaria vectors and vertebrates, including the recent paper in Nature on introgression in Darwin’s finches. For example, in the Discussion we highlight literature on yeast genome evolution, in which the authors infer recent hybridizations based on a genome analysis of introgression in *S. cerevisiae.* Introgressed regions from various other yeast species can be nearly identical to those in the donor genomes, so hybridisation must have occurred relatively recently. We draw a similar conclusion for *Albugo*, and in addition, we estimate the dates when recombination or hybridisation has occurred in our study system.

*2) The data suggest that 25% of the genome of a race is introgressed from a genome from a different host race. With this much introgression, and assuming that this study did not find exceptional cases, but rather a representative sample from the wild, one wonders about the genetic architecture of host-range determination. If host-ranges are determined by multiple loci, one would except that recombination will quickly lead to super genotypes, with very wide host ranges. Apparently this did not happen; maybe recombination cannot change this. This would be the case if they are sitting in non-recombining gene clusters, or a single locus (with multiple alleles) is responsible for host ranges. We would like to see some discussion about this*.

We are here asked to consider the genetic architecture of host-range determination and why recombination does not lead to “super genotypes”, with very wide host ranges. In the Introduction we state: “The adaptive evolution of, for example, new effectors that enable more efficient exploitation of one host species, increases the risk of detection in other host species by triggering their immune system (32)”. In the Discussion we also state: “If host-ranges are determined by multiple loci […] an adaptive advantage assuming that the effector is not recognised.”

*3) The authors use the term “immunosuppression”. To an animal immunologist, this implies that the study included the assessment of immunological parameters, which is not the case. Rather, you observe in experimental infections that a first infection facilitates infection with a second race of the pathogen. We suggest rephrasing this, so as to not confuse animal immunologists*.

We have taken on the reviewer’s comment and entirely removed all instances of this word from the manuscript. However, suppression of innate immunity of both *Arabidopsis thaliana* and *Brassica juncea* by *Albugo candida* was first described by Cooper et al., 2008a (cited in the manuscript). The authors document suppression of plant immunity by *A. candida* to another *A. candida* race and to a downy mildew species. Cooper et al. selected hosts for study because they had distinct resistance gene pathways and they concluded, “*Albugo candida* subsp. Arabidopsis and its defense suppression capability represent an important and fascinating complement for downy mildew research in *Arabidopsis thaliana*”.